# Pharmacogenomics Variability of Lipid-Lowering Therapies in Familial Hypercholesterolemia

**DOI:** 10.3390/jpm11090877

**Published:** 2021-08-31

**Authors:** Nagham N. Hindi, Jamil Alenbawi, Georges Nemer

**Affiliations:** 1Division of Genomics and Translational Biomedicine, College of Health and Life Sciences, Hamad Bin Khalifa University, Doha P.O. Box 34110, Qatar; nhindi@hbku.edu.qa (N.N.H.); jalenbawi@hbku.edu.qa (J.A.); 2Department of Biochemistry and Molecular Genetics, Faculty of Medicine, American University of Beirut, Beirut DTS-434, Lebanon

**Keywords:** familial hypercholesterolemia, pharmacogenomics, PCSK9 inhibitors, statins, ezetimibe, novel lipid-lowering therapy

## Abstract

The exponential expansion of genomic data coupled with the lack of appropriate clinical categorization of the variants is posing a major challenge to conventional medications for many common and rare diseases. To narrow this gap and achieve the goals of personalized medicine, a collaborative effort should be made to characterize the genomic variants functionally and clinically with a massive global genomic sequencing of “healthy” subjects from several ethnicities. Familial-based clustered diseases with homogenous genetic backgrounds are amongst the most beneficial tools to help address this challenge. This review will discuss the diagnosis, management, and clinical monitoring of familial hypercholesterolemia patients from a wide angle to cover both the genetic mutations underlying the phenotype, and the pharmacogenomic traits unveiled by the conventional and novel therapeutic approaches. Achieving a drug-related interactive genomic map will potentially benefit populations at risk across the globe who suffer from dyslipidemia.

## 1. Introduction

Familial hypercholesterolemia (FH) is the most common and first acquired pathology of lipoprotein metabolism to be characterized genetically and clinically [1]. It is identified by elevated levels of absolute low-density lipoprotein cholesterol (LDL-C) in the blood, early-onset of atherosclerotic cardiovascular diseases (ASCVD), fat accumulation in external tissues, and tendon and cutaneous xanthomas [2]. Globally, heterozygous FH afflicts around one in 250–500 individuals, with a more noticeable predominence in special communities, including the Christian Lebanese, French-Canadian, Finnish, and Afrikaner [3]. Medical symptoms of the severe phenotype, a homozygous FH, initiates at the early stages of childhood with a predicted incidence of one in a million. The intensity of FH complications such as coronary ostium and aortic root predominantly rely on total LDL-C levels [4,5]. Clinical examination of FH can be verified based on premature cardiovascular diseases (CVD), physical marks, and a history of raised cholesterol levels. In addition to the serum lipids analysis, various systemic diagnostic guidelines suggest cascade genomic examining to detect FH and confirm the polymorphisms in family members up to the third degree, including the Dutch Lipid Clinic Network—Make Early Diagnosis to Prevent Early Deaths (Dutch-MEDPED) [6]. Genetic testing could facilitate early recognition and treatment of undiagnosed, untreated FH patients and is known to provide a better prognosis of the disease.

Inherited disease-causative impairments of the low-density lipoprotein-receptor gene, (*LDLR*) present in 70–90% of subjects, and, less commonly, the apolipoprotein B gene (*APOB*), as well as proprotein convertase subtilin/Kexin, member nine genes (*PCSK9*) have been linked to raised lipoprotein cholesterol in FH (190–400 mg/dL) [3]. Additional genes encoding the LDLR-adaptor protein 1 (*LDLRAP1*) and apolipoprotein E (*APOE*) can infrequently correlate with cholesterol homeostasis and promote the development of autosomal recessive FH [4]. *APOB* and *APOE* genes are responsible for encoding ApoB-100 and ApoB-48 isoforms as well as ApoE, respectively, which are the elemental apolipoproteins of the LDL-C and are the protein ligands of LDLR. *PCSK9* gene encodes member 9 of the PCSK family that involves the lysosomal degeneration and coordination of LDLR. The LDLRAP1 protein encoded by the *LDLRAP1* gene has a phosphotyrosine binding domain that interacts and harmonizes the LDLR activity. The physiological uptake and catabolism of fats are essentially mediated by hepatic LDLR, which is encoded via the *LDLR* gene [3,4]. Interestingly, the number of variations in *LDLR* and associated genes related to the clinical manifestations of FH is uniformly rising.

For a long time, there was an apparent focus on investigating *LDLR* variants to recognize the impact on the medical, biochemical, and pathological phenotypes of FH monogenic dysfunctions. It is noteworthy that the significant phenotypic diversity of lipids and coronary artery disorders depends on the nature of FH genetic defects. These defects are modulated, however, by various genetic and epigenetic factors and, thus, various pathological genotypes can differentially impact the circulating levels of LDL-C [7,8]. For instance, a nonsense variant in the *LDLR* (c.2043 C>A, p. cys681X) was predominantly combined with familial hypercholesterolemia in nearly 82% of Lebanese cases. This Lebanese allele leads to a *LDLR* loss-of-function (null) defect and attenuates hepatic metabolism and removal of LDL-C and is believed to lead to a very severe phenotype [9]. Paradoxically, the mutation is a founder mutation in the Lebanese population and was encountered in Lebanese individuals with normal cholesterol levels. This indicates the presence of unrecognized variants and/or an epigenetic signature that counters the effects of the deleterious LDLR mutation in these cases [10]. Consequently, genetic diagnostic screening of disease-causative mutations, considered the gold standard for FH detection, is not enough, but should be coupled with whole-genome sequencing and/or methylation analysis to further stratify affected members within familial cases.

Despite the prevalence of FH and the significance of early determination and management of the condition, only 15–20% of FH subjects are diagnosed by medical examination. Untreated patients with heterozygous FH have a nearly 20-fold higher raised incidence of premature coronary artery disease relative to cases without FH [11]. Coronary artery disease and heart attacks restrict coronary blood flow, causing the pumping chamber to enlarge, widen, and attenuate. Ultimately, this damage will lead to ischemic cardiomyopathy, potentially reducing the ability of cardiomyocytes to pump blood [12]. The earliest clinical mark of the disease regularly happens throughout the third decade of growth, particularly in severe cases with LDLR-negative mutational status, such as fatal myocardial infarction [5]. Appropriate identification and management could control the lipid levels and limit the life-threatening complications of FH. A current examination documented that early introduction of lipid-lowering medicine throughout childhood and adolescents in cases with FH can ameliorate the pathological progression and reduce the incidence of ASCVD, explaining the considerable advantage of immediate FH treatment [13]. Numerous studies have revealed that only 10% reached the recommended cholesterol levels even though most patients receive the maximum tolerated cholesterol-modulating drugs [14].

The pharmacological variation among FH patients has been linked to the genomic single nucleotide polymorphisms (SNPs) of genes associated with cholesterol catabolism and biosynthesis [15]. In extreme cases of FH, genetic screening could potentially be used to examine the response variability and, therefore, to effectively personalize the therapeutic care plan for the anti-lipid interventions and CVD risk preventions. Under this scenario, this review will discuss the management of familial hypercholesterolemia with the standard and innovative therapeutic strategies from the prospect of pharmacogenomics and its link to the causative genetic mutations underlying the phenotype.

## 2. FH Management

The main goal of FH therapy is to reduce relative LDL-C by more than 50% or to lower LDL-C to 100 mg/dL in adults without ASCVD. For FH subjects with ASCVD or major CVD risk, the 2019 ESC/EAS guidelines recommend a more than 50% reduction of LDL-C or less than 55 mg/dL of absolute LDL-C [6]. Therapeutic lifestyle modifications such as restricted diet, regular physical training, limiting alcohol intake, and smoking cessation are all fundamental in the controlling of FH [16]. In addition, patients should be counseled to maintain healthy blood sugar, blood pressure, and weight. Despite the paramount importance of non-pharmacological management in all FH patients, optimizing cholesterol levels and preventing CVD are hardly achieved without pharmacological interventions [5].

Currently, β-hydroxy-β-methylglutaryl coenzyme A reductase (HMGCR) inhibitors at the highest tolerable dose are strongly recommended to be initiated immediately at diagnosis in all FH adults. Monotherapy, daily doses of the aggressive statins, and HMGCR inhibitors, including atorvastatin 40–80 mg and rosuvastatin 20–40 mg orally per day, are expected to decrease LDL-C approximately 50–60%, as reported in various cholesterol-lowering studies [11,14]. When the target fails to be achieved, stepwise intensification of anti-lipid medications should be considered. Ezetimibe, a cholesterol uptake blocker, can decrease the LDL-C by nearly 25% and is recommended as an adjunctive second-line treatment [6,11].

PCSK9-based medications are a great breakthrough in FH pharmacotherapy, with a reduction in LDL-C ranging from 25% to 30%. The anti-PCSK9 monoclonal antibodies, evolocumab and alirocumab, should be initiated if maximum intense statins and ezetimibe fail to sufficiently control the lipid profile in FH cases with a major risk of cardiomyopathies [6]. In 2013, the Food and Drug Administration (FDA) approved using a microsomal triglyceride transfer (MTP) inhibitor, lomitapide, and an oligonucleotide agent, mipomersen, as add-on medications to the classic anti-lipids in adults with the severe FH phenotype. Lipoprotein apheresis should be offered for individuals with either homozygous FH or severe-uncontrolled heterozygous FH, which may reduce 60–80% of LDL-C [6]. The cholesterol metabolism pathway and pharmacological targets for classical and novel lipid-lowering therapies are schematically summarized in Figure 1.

For familial genotyped cases, the current therapeutic protocols suggest maintaining LDL-C below 135 mg/dL for children aged 10 or older and reducing LDL-C by 50% for younger ages. A healthy lifestyle could achieve this goal in conjugation with a mild statin regimen, subject to dosage increase, to reach the targeted-cholesterol levels [6,16]. Nevertheless, various FH cases are still not controlled despite aggressive therapeutic interventions. Contemporary information from the International Pediatric FH Register revealed that 23% of heterozygous FH children on statins had not reached the targeted LDL-C goal (below 135 mg/dL) [17].

The wide variation of cholesterol-lowering therapies in terms of the potential adverse effect and therapeutic response is one of the main challenges in clinical practice, especially in FH patients. In addition to clinical and environmental factors, including race, gender, age, smoking, and adverse consequences, genomic phenotypes of *LDLR*, *APOB*, and *PCSK9* can potentially modulate the sensitivity of anti-lipids. Over the previous decade, many pharmacogenomics and genome-wide association studies (GWASs) have recognized numerous genetic variations that can affect the therapeutic potency (anti-lipid pharmacodynamics), drug absorption, metabolism, excretion (anti-lipid pharmacokinetics), and anti-lipid toxicity pathways [3,18]. Accordingly, therapeutic efficiency and safety and patient quality of life could be promoted through personalized genomic examination, which is designed to predict the therapeutic response of FH management.

## 3. Pharmacogenomics of Statin in FH

The primary and secondary prevention of CVD and the cornerstone medication in patients with FH are via HMGCR inhibitors [5,6]. Statins could potentially decrease the plasma levels of atherosclerotic LDL-C via competitively inhibiting the HMGCR (Figure 1) [11]. The inhibition of this protein reduces the hepatic synthesis of cholesterol and, thereby, enhances LDLR production. Subsequently, the elevated expression of LDLR on the hepatocytic membrane will increase the cellular uptake of cholesterol from the bloodstream, mainly by the liver. Furthermore, the secretion of ApoB-containing lipoproteins, LDL, and very-low-density lipoprotein (VLDL), as well as triglycerides from hepatocytes, may also be lowered via statins [11]. The lifelong overburden of high cholesterol makes patients with FH highly susceptible to the risk of CVD and significantly reduces their life expectancy [2]. Although statins robustly diminish cholesterol in addition to CVD morbidity and mortality by 20–30% in normal individuals, their efficacy is predominantly weaker in FH subjects [5]. Genetic variations combined with non-adherence due to statin myotoxicity or hepatotoxicity may cause pharmacological variability among patients. We will divide the variants according to the effect they have on either the pharmacodynamics or the pharmacokinetics of these drugs.

### 3.1. SNPs Linked to Pharmacodynamics of Statins in FH

The hepatocyte endocytosis of lipoproteins is mediated mainly by LDLR in addition to other processing associated proteins, including PCSK9, APOE, and LDLRAP1. SNPs in the *LDLR* could selectively reshape the anti-lipids therapeutic outcome and the incidence of FH and coronary artery conditions. Therefore, the pharmacogenetic analysis principally concentrates on discovering these mutations, as reviewed in Table 1 [19,20,21,22,23,24]. Polisecki and colleagues observed a strong association between the serum-baseline cholesterols and statin efficacy in terms of coronary artery disease risk in FH patients carrying an *LDLR* polymorphism (rs1433099, c.44857C>T) [25]. The 3′-untranslated region (3′-UTR) of LDLR has been found to play a basic role in the anti-lipids mediated-LDL-C reduction through stabilizing the LDLR mRNA. Polymorphisms at the *3-UTR* loci have been linked to lipid baselines, LDLR activity, and CVD [26]. Interestingly, subjects with mixed *LDLR* and *HMGCR* haplotypes have more prominent attenuations in optimizing desired cholesterols than those carrying a single *LDLR* mutation [27]. The cholesterol-lowering potency of pravastatin has also been modulated by another *LDLR* genetic defect (rs5925, c.2052T>C) [28]. Recently, *LDLR* stop-gained pathogenic variants (c.2027delG, p. Gly676Alafs*33) have been discovered to be correlated with anti-lipid efficacies including statins, ezetimibe, and clinical manifestations of FH [29].

Several investigations have noticed that the physiological effect of lipid-modifying drugs, especially statin, and the lipid profile in heterozygous FH subjects is affected by the presence of LDLR variants and the type of mutation. Individuals carrying a null mutation (deletions result in a frameshift and premature stop codon) were observed to have diminished LDL-C responses with elevated cholesterols compared to patients carrying a defective (non-frameshift small insertions or deletions) or without mutation [30,31,32,33,34].

**Table 1 jpm-11-00877-t001:** Pharmacogenomics variations associated with statin response in familial hypercholesterolemia patients.

Gene	Significant Mutation *	Patients	Population	Sample Size	Treatment and Daily Dose	Clinical Findings	Author, Year (References)
*LDLR*	FH1 (C206G) & FH2 (G408A)	Het-FH	Afrikaners	20	Simvastatin 40 mg	TC reduction is higher in patients with FH2 than FH1	Jeenah et al., 1993 [19]
*LDLR*	C660X, D147H, & 652delGGT	Het-FH	Israeli	64	Fluvastatin 40 mg	Reduction of LDL-C, apoA, and elevation of HDL-C depend variously on *LDLR* mutations	Leitersdorf et al., 1993 [23]
*APOE*	E2,3, & 4 alleles	Het-FH	Canadian	49	Lovastatin 80 mg	Statin sensitive is higher in men with E4 than E3 or E2 or women with any *APOE* phenotype	Carmena et al., 1993 [35]
*LDLR*	FH_TONAMI-1_ (Del exon15) &FH_KANAZAWA_ (C665T)	Het-FH	Japanese	12	Pravastatin & cholestyramine	LDL-C reduction is higher in patients with FH_KANAZAWA_ than FH1 FH_TONAMI-1_	Kajinami et al., 1998 [20]
*LDLR*	W66G, C646Y, & deletion>15 kb	Het-FH	Canadian	63	Simvastatin 20 mg	LDL-C reduction is higher in patients with C646Y & deletion > 15 kb than W66G	Couture et al., 1998 [21]
*LDLR*	Severe and mild LDLR	Het-FH	British	42	Simvastatin + bile acid sequestrant	LDL-C is higher in patients with severe than mild mutation	Sun et al., (1998) [33]
*LDLR*	Null and defective *LDLR*	Het-FH	British	109	Simvastatin	LDL-C reduction is higher in patients with defective than null mutation	Heath et al., (1999) [31]
*LDLR*	AvaII (rs5925T>C), HincII (rs688C>T), & PvuII (rs2569542A>G)	Het-FH	Brazilian	55	Fluvastatin 40–80 mg	LDL-C, TC, & ApoB reduction is higher in patients with AvaII & PvuII than HincII	Salazar et al., 2000 [22]
*LDLR*	Null and defective *LDLR*	FH	Spanish	55	Simvastatin 20 mg	Low HDL-C & poor statin response are higher in patients with defective than null mutations	Chaves et al., (2001) [32]
*APOE*	E4 allele	Het-FH	British	19	Atorvastatin 10 mg + bile acid sequestrant	Poor statins response is high in patients with E4 phenotype	O’Neill et al., 2001 [36]
*LDLR*	Null and defective *LDLR*	Het-FH	Canadian	63	Atorvastatin 20 mg	LDL-C reduction is higher in patients with null than defective mutation	Vohl et al., (2002) [37]
*LDLR*	G1775A, G1646A, & C858A	Het-FH	Greek	49	Atorvastatin 20 mg	LDL-C & ApoB reduction is higher in patients with G1775A than G1646A & C858A	Miltiadous et al., 2005 [24]
*MTP*	c.493 GT	Het-FH	Spanish	222	Atorvastatin 20 mg	High reduction of TG in men and low reduction of VLDL & TG in women with c.493 GT allele	García-Garc ía et al., 2005 [38]
*CETP*	−867 and Ex14/I405V	Het-FH	Israeli	76	Fluvastatin 40 mg	LDL-C reduction is high among *CETP* & *MDR1* mutants	Bercovich et al., 2006 [39]
*MDR1*	c.(G2677T) and c.(C3435T)						
*LDLR*	Null and defective *LDLR*	Het-FH	Spanish	811	Simvastatin or atorvastatin 80 mg ± bile acid sequestrant	PCVD & TC is higher in patients with null than defective mutations	Alonso et al., 2008 [40]
*ABCG2*	rs2231142	FH	Chinese	386	Rosuvastatin 10 mg	High LDL-C reduction among patients with AA genotype	Hu et al., 2010 [41]
*LDLR*	Null and defective *LDLR*	FH	Spanish	387	Maximum statin doses ** + ezetimibe 10 mg	Poor LLT response & high PCVD in patients with null than defective mutations	Mata et al. (2011) [42]
*LDLR*	W556R	Twins with Hom-FH and parents with Het-FH (one family)	Turkish	4	Simvastatin 40 mg + ezetimibe 10 mg or LDL apheresi	Hom-FH have a low LDL-C reduction and high statin resistance, but Het-FH respond to statin with 60% LDL-C reduction	Schaefer et al., 2012 [43]
*CYP3A4*	rs2740574	FH	Chilean	142	Atorvastatin 10 mg	High statin sensitivity among patients with *CYP3A4* mutations	Rosales et al., 2012 [44]
*ANRIL*	rs1333049	FH with CVD	Pakistani	611	Atorvastatin 10, 20 or 40 mg	High LDL-C, TC, & TG reduction in patients with CC genotype	Ahmed et al., 2013 [45]
*LDLR*	Null (W66G) and defective (C646Y) *LDLR*	Het-FH	Brazilian	156	Atorvastatin 10, 20 or 40 mg	LDL-C reduction is more in patients with defective than with null mutation	Santos et al., 2014 [30]
*POR*	rs1057868	FH	Greek	105	Atorvastatin 10, 20 and 40 mg	High LDL-C & TC reduction in patients with 1/1 genotype	Drogari et al., 2014 [46]
*MYLIP*	rs9370867	Het-FH	Brazilian	156	Atorvastatin 10–80 mg ± ezetimibe 10 mg	High LDL-C reduction in patients with AA genotype	Santos et al., 2014 [47]
*PSCK9*	E32K	Hom-FH	Japanese	1055	Atorvastatin 80 mg & ezetimibe 10 mg	*PSCK9* gain-of-function variants significantly worsen *LDLR* phenotype and decrease LDL-C reduction	Mabuchi, et al., 2014 [48]
*LDLR*	Double allele						
*LDLR*	Null and defective *LDLR*	FH	Spanish	4132	Maximum statin doses ** + ezetimibe 10 mg	Poor LLT response & CVD events are higher in null than in defective mutation	Perez de Isla et al., 2016 [14]
*LDLR*	p.(Cys155Gly)	Hom-FH	Belgian	8	Atorvastatin 80 mg, ezetimibe 10 mg, cholestyramine	LLT efficacy is attenuated in patients with nonsense *LDLR* mutations	Sanna et al., 2016 [34]
*HMGCR*	rs3846662	Het-FH	French Canadian	106	Statin + LLTΩ	Poor statin response among *HMGCR* mutants	Leduc et al., 2016 [49]
*LDLR*	W87G, C368Y, T726I, G2fsX214, D47N, N97H, E101K, C216fsX, L582P, C667Y, & LDLR-17-18 del	Het-FH	American and Canadian	139	Atorvastatin 40/80 mg, rosuvastatin 20/40 mg or simvastatin 40/80 mg, + Bococizumab 0.25, 1, 3, or 6 mg/kg	Bococizumab effecacy is higher than statin in reducing LDL-C across *LDLR* & *APOB* variants	Fazio et al., 2018 [50]
*APOB*	R3527Q						
*LDLR*	Het-*LDLR* mutation	FH	Spanish	22	Maximum statin doses ** ± ezetimibe 10 mg	LDL-C reduction is higher in patients with p.(Leu167del) mutation than LDLR	Bea et al., 2019 [51]
*APOE*	p.(Leu167del)						
*LDLR* *SLCO1B1* *ABCB11* *CYP3A5*	*rs28941776* c.(521T>C; SLCO1B1*5) & c.(388A>G; SLCO1B1*1B) rs2287622 CYP3A5*3	FH	Caucasian	1	Rosuvastatin 40 mg & ezetimibe 10 mg	Loss-of-function mutations enhance statin myotoxicity and delay its response	Dagli-Hernandez et al., 2021 [52]
*LDLR*	c.(2027delG), p. (Gly676Alafs*33)	FH (2 families)	Saudi	12	Statin + ezetimibe	Clinical manifestations and poor LLT response depend on LDLR variants	Awan et al., 2021 [29]

* Borderline significance (*p* < 0.05). ** Maximum tolerated dose of statin: simvastatin 80 mg, pravastatin 40 mg, lovastatin 80 mg, fluvastatin 80 mg, atorvastatin 80 mg, or rosuvastatin 20–40 mg. Ω LLT including ezetimibe, fibrates, statins, lomitapide, PCSK9 inhibitors, bile acid sequestrants (e.g., cholestyramine and colestipol), or niacin. Abbreviations: LLT; lipid-lowering therapies; PCVD, premature cardiovascular diseases; FH, familial hypercholesterolemia; Het-FH, patients with heterozygous FH; Hom-FH, patients with homozygous FH; ApoB, Apolipoprotein B protein; HDL-C, High-density lipoprotein cholesterol; LDL-C, low-density lipoprotein cholesterol; TC, total cholesterol; TG, triglyceride; *LDLR,* Low-density lipoprotein receptor; *APOB*, Apolipoprotein B; *ABCG2,* atp-binding cassette, subfamily g, member 2; *MDR1,* multidrug resistance mutation 1; *CYP3A4,* Cytochrome P450, family 3, subfamily A, member 4; *ANRIL,* antisense non-coding RNA in the INK4 locus; *POR*, Cytochrome P450 Oxidoreductase; *MYLIP,* Myosin Regulatory Light Chain Interacting Protein; *HMGCR,* β-hydroxy-β-methylglutaryl Coenzyme A Reductase; *E,* Epsilon; *SLCO1B1,* solute carrier organic anion transporter 1B1.

In addition, FH patients with a null mutation in the *LDLR* gene were identified as having a higher prevalence of CVD than those with a defective mutation [14,40,42,53]. Although these individuals at major risk of CVD are on aggressive anti-lipid regimens, most of them did not achieve the therapeutic goals of LDL-C [37,42]. On the contrary, a study by Vohl and colleagues found that the proportion of patients who achieved LDL-C targets was higher in the null mutants than in the defective mutants [37]. Schaefer et al. have confirmed that *LDLR* p.W556R SNP in homozygote FH patients lead to HMGCR blockers resistance but can obtain a 15% decrease of LDL-C by ezetimibe treatment. Conversely, the same *LDLR* mutation in patients with heterozygote FH can decrease 60% of cholesterols under a combination of ezetimibe and simvastatin [43]. These outcomes suggest that altering the *LDLR* should be a new pharmacological target in controlling FH.

Pharmacogenomic assays have shown that low-activity variants of *HMGCR*, which encode the cholesterol synthesis speed-limiting factor, can restrict the therapeutic potency of HMGCR blockers depending on the patients’ gender. For instance, the *HMGCR* polymorphism, rs3846662, selectively modulates women’s sensitivity to statin treatments [49]. Variations in the encoding genes of ApoA molecules and lipoprotein (A) (LPA), have been believed to constrain LDL-C response to statins and intensify coronary artery disorders [54]. Several GWAS studies have proved an association between *PCSK9* polymorphisms and statin efficacy. The rs17111584 C allele in *PCSK9* decreased the rosuvastatin efficacy [55], while the rs11599147 polymorphism was linked to elevated anti-lipid response [56]. A polymorphism in the WD repeat domain 52 (*WDR52*, rs13064411A>G) can indirectly reduce the LDLR response to statins. This mutation is associated with statin-induced elevation of PCSK9 levels that accelerate the degradation of LDLR, resulting in elevated total cholesterol levels [57]. The myosin regulatory light chain interaction protein (MYLIP) is responsible for regulating the LDLR function in cellular lipid uptake. A study noted that heterozygous FH patients with the *MYLIP* rs9370867 allele respond differently to statin therapy with ezetimibe based on the mutation type. After a year of treatment, the recommended cholesterol levels could be achieved in FH patients with no mutations but not in those with defective and null phenotypes [47]. All in all, the results from various studies point out to an essential role for the *LDLR* mutation type in predicting response to statins but also to a preponderant role to genes involved in LDLR regulation as potential modifiers to this response.

### 3.2. SNPs Linked to Pharmacokinetics and Pharmacotoxicity of Statins in FH

Gene polymorphisms associated with pharmacokinetic and toxicokinetic may greatly contribute to the attenuated response to HMGCR inhibitors. Figure 2 reviews candidate pharmacokinetic modulator genes involved in the distribution, metabolism, and elimination of anti-lipids. The hepatic absorption of statins is mediated primarily by the solute carrier organic anion transporter 1B1 (encoded via *SLCO1B1*). Loss-of-function *SLCO1B1**5 (c.521T>C) and *SLCO1B1**1B (c.388A>G) SNPs remarkably diminish plasma LDL-C transporting to the liver and raise the systemic exposure to statin [52]. This results in a greater incidence of rhabdomyolysis risk and a negligible cholesterol optimizing effect. However, many genomic examinations have failed to connect polymorphisms in *SLCO1B1* and statin-mediated cholesterol modifying effect or myotoxicity [46,58,59]. Mutations in the ATP-binding cassette transmembrane mediator (ABC) have been significantly correlated with impaired efflux of statins as well as cholesterols from cells. The attenuated activity of *ABCA1, ABCA11,* and *ABCG2* was found to reduce the excretion of statin and increase its intrahepatic levels, thereby increasing hepatotoxicity as well as myopathy to statin adverse consequences [41,52].

Polymorphisms of the *APOE* gene (*E2*, *E3*, and *E4*) in humans have diverse effects on removing apolipoproteins from circulation. Inhibition of the HMGCR enzyme is most evident in lipoproteins with the *APOE4* allele, which is very efficiently removed from the blood. Various clinical trials have identified that *APOE4* variants are often accompanied by elevated fats absorption, enhancing the endogenous cholesterol catabolism. Consequently, these variations may alleviate the modulation of atherogenic LDL-C in response to HMGCR inhibitors [36]. Accordingly, patients should be counseled to sustain a healthy diet or combine statins with absorption inhibitors. On the other hand, patients carrying the *E2* isoform and the *APOE* (p. Leu167del) mutation respond more efficiently to the conventional anti-lipid therapy, consistent with the poor plasma clearance that enhances the HMGCR synthesis [35,36,51]. Generally, *APOE* variations are not directly targeting the statin pharmacokinetic pathway. However, they are affecting the expression of plasma lipids and thus altering the pharmacodynamic responses of statins.

Variations of cytochrome P450 (*CYP450*) may exceedingly impact anti-lipids metabolism and, thus, result in a diversity of LDL-C response and adverse consequences among FH patients. The byproduct of these enzymes has a principal role in inhibiting the HMGR protein, indirectly promoting statin effectiveness. Therefore, nonfunctional *CYP3A5*3* mutations were reported to lower the rosuvastatin efficacy in decreasing the LDL-C [52]. On the contrary, Rosales et al. have reported that *CYP3A4* polymorphism rs2740574 (-290A>G) enhances atorvastatin therapeutic response in subjects with FH [44]. The activity of CYP3A is chiefly controlled through the electron transferring function of cytochrome P450 oxidoreductase (POR) from NADPH. *POR*28* rs1057868C>T SNP has been combined with raised functionality of *CYP3A* in the FH cohort, explaining the diverse therapeutic responses to statin [46]. Nonetheless, many studies found that mutations in *CYP450* genes are not linked to anti-lipids intolerance [44].

Hepatic metabolism of various compounds, including statins, can be mediated through the metabolic function of N-acetyltransferase type 2 (NAT2). A mutation in this enzyme can either enhance or delay physiological metabolism. A considerable variation in the statin pharmacokinetics was reported in *NAT2-rs1208* polymorphism carriers [60]. Interestingly, a wide pharmacogenomic investigation revealed an association between the *NAT2*1* SNP and a significant LDL-C decrease in response to simvastatin [61]. These findings could be potentially used to guide medical decision-makers to improve the therapeutic plan for FH patients. Nevertheless, the consequence of *NAT2* mutations on anti-lipid pharmacokinetics has not yet been determined in FH.

The Bioavailability of statins has also been linked to other genes, including P-glycoprotein drug transporter (MDR1). *MDPR1* regulates the uptake, distribution, and removal of statin from renal, hepatic, and intestinal cells. Certain polymorphisms in the *MDR1* gene, such as G2677T and C3435T, can modulate statins transportation and, thus, enhance the cholesterol regulatory effect [39]. Mutations have also been noted in other pharmacokinetic modulator genes, such as *ANRIL, CETP,* and *CYP2C9,* that could contribute to the interindividual variations of FH therapy, summarized in Table 1 [39,45,46]. However, the impact of the identified variants on statin-mediated reduction of LDL-C compared to the *LDLR* polymorphisms is insignificant. None of them showed any significant relationship with the clinical outcomes.

## 4. Pharmacogenomics of Non-Statin Lipid-Lowering Therapies in FH

Multiple non-statin therapies effectively control cholesterol levels and could be prescribed as mono- or combined therapy in FH patients, including ezetimibe, PCSK9 inhibitors, mipomersen, and lomitapide. The latest recommendations advise intensifying the management with non-statin medicines on top of maximum statins for resistant or non-adherent statin-induced muscle pain [6]. To date, many biogenetic analyzes have been performed to examine these factors, as summarized in Table 2. However, further pharmacogenomic investigations are required to comprehensively understand the clinical response in the FH population.

### 4.1. Ezetimibe

Modulation of intestinal cholesterol absorption by ezetimibe 10 mg orally a day is another targeted pathway to further reduce the cholesterol levels in FH patients. It targets the cholesterol transporter Niemann-Pick C1-like one protein (encoded by *NPC1L1*) in the liver and small intestine, thus inhibiting the endogenous cholesterol synthesis and upregulating the LDLR expression. Several genetic mutations involved in lipid transfer can modulate the pharmacodynamic effects of ezetimibe treatment [29]. For instance, ezetimibe’s reduction of cholesterol absorption was elevated in patients with mutations in the sterol regulatory binding protein 1 gene (*SREBP-1c*) [62]. Furthermore, the risk of developing ASCVD was significantly associated with a lower response to ezetimibe caused by a polymorphism in the *NPC1L1* gene (rs55837134) and statins by *HMGCR* mutations [63]. The ATP-binding cassette, subfamily G, member 5 (*ABCG5*) or 8 (*ABCG8*), plays an essential role in the intestinal secretion of cholesterol. A patient with a novel heterozygous *ABCG5* mutation (c.203A>T; p. Ile68Asn) manifested great sensitivity to ezetimibe and resisted the statins medication [64]. Cases such as this support the consideration of ezetimibe use for all patients with hypercholesterolemia who are resistant to HMGCR inhibitors.

**Table 2 jpm-11-00877-t002:** Pharmacogenomics variations associated with non-statin & novel LLT responses in familial hypercholesterolemia patients.

Gene	Significant Mutation *	Patients	Population	Sample Size	Treatment and Daily Dose	Clinical Findings	Author, Year (References)
**Non-statin Lipid-Lowering Therapies**							
*LDLR*	Defective and negative *LDLR*	Hom-FH	South African	8	Evolocumab 140–420 mg every 2–4 weeks for 3 months	Evolocumab is reducing LDL-C in LDLR-defective but not in negative cases	Stein et al., 2013 [65]
*LDLR*	Defective and negative *LDLR*	Hom-FH	10 countries **	50	Evolocumab 420 mg every 4 weeks for 3 months	Evolocumab responses is LDLR-genotype dependent with higher sensitivity in *LDLR*-defective patients	Raal et al., 2015 [66]
*PCSK9*	rs28942111 (S127R) rs28942112 (F216L)	Het-FH	27 countries **	2341	Statin maximum dose + LLT & alirocumab 150 mg/2 weeks for 78 weeks	Alirocumab is significantly reducing LDL-C in *PCSK9* gain-of-function variants	Robinson et al., 2015 [67]
*LDLR*	c.(1646G <A)	Hom-FH	Italian	15	Simvastatin 40 mg, ezetimibe 10 mg, & lomitapide 5–60 mg	Lomitapide is significantly and safely decreasing the cholesterol levels	D’Erasmo et al., 2017 [68]
*LDLRAP1*	c.(432_433insA)						
*LDLR*	Defective and negative *LDLR*	Hom-FH	South African	22	Mivastatin and evolocumab	Evolocumab is effective in defective- and not in negative-*LDLR* variants	Thedrez et al., 2017 [15]
*APOB*	R3500Q (rs5742904)	Het-FH	Caucasian	1	Atorvastatin 80 mg, ezetimibe 10 mg, lomitapide, & evolocumab 140 mg	*ApoB* defect is enhancing LDL-C reduction	Andersen et al., 2017 [69]
*LDLRAP1*	c.136 C > T (406)	AR-FH	German	1	Atorvastatin 80 mg, ezetimibe 10 mg, lomitapide, & evolocumab 140 mg	Evolocumab is reducing LDL-C by 37% among *LDLRAP1* mutants	Fahy et al., 2017 [70]
*LDLR*	Two null alleles	Hom-FH	American	9	LLTΩ + Evolocumab 420 mg/4 weeks	Evinacumab is controlling cholesterol independently of *LDLR* variants	Gaudet et al., 2017 [71]
*LDLR*	c.2043C.A, p. (Cys 681A)	Het-FH	Lebanese American	1	Rosuvastatin, ezetimibe, & evolocumab 140 mg/2 weeks for 2 months, then alirocumab 150 mg/ 2 weeks	Alirocumab efficacy is higher than evolocumab & standard LLT	Doyle et al., 2018 [72]
*LDLR*	p.(Thr434Arg)	Hom-FH	Spanish	2	LLTΩ & lomitapide 20–40 mg	Lomitapide is potent and safe as adjunct therapy	Real et al., 2018 [73]
*LDLRAP1*	c.1A > G	AR-FH	Spanish	3	Atorvastatin, ezetimibe, & evolocumab	Evolocumab effecacy is lower in *LDLR & LDLRAP1* variants	Rodríguez-Jiménez et al., 2019 [74]
*LDLR*	p.(Cys352Ser) & p.(Asn825Lys)						
*LDLR PCSK9 APOB*	p.(Trp87Gly), p.(Gln254Pro), & p.(Ala627Profs*38)	Hom-FH	Chinese	9	LLTΩ + evinacumab 250 mg	Evinacumab is controlling cholesterol independently of *LDLR* variants	Banerjee et al., 2019 [75]
*PSCK9*	c.137 G>T, p.(Arg46Leu)	Hom-FH	Caucasian	3	LLTΩ + Evolocumab 420 mg/ 4 weeks	Evolocumab is strongly reducing LDL-C and CVD in *PSCK9* loss-of-function mutants	Bayonaet al., 2020 [76]
*LDLR*	c. 902A>G, p.(Asp301Gly)						
*LDLR*	p.(Pro685Leu)	Hom-FH	Indian	1	LLTΩ & evolocumab 420 mg/4 weeks + lomitapide 5–60 mg	Lomitapide is powerfully reducing lipid profile and CVD risk	Velvet et al., 2020 [77]
*LDLR*	Null mutation in both alleles	Hom-FH	13 countries **	69	Atorvastatin 80 mg, ezetimibe 10 mg, lomitapide, & alirocumab 150 mg/2 weeks for 12 weeks	Alirocumab is effective in controlling the lipid profile	Blom et al., 2020 [78]
*LDLR*	c.2027delG, p.(G676Afs*33)	Hom-FH	Saudi	2	Rosuvastatin 40 mg, ezetimibe 10 mg, evolocumab 420 mg/month, & lomitapide 5–40 mg	Lomitapide is robustly reducing cholesterol and CVD events	Mahzari et al., 2021 [79]
**Novel Lipid-Lowering Therapies**							
*LDLR* *APOB* *PCSK9*	Deficient and defective	Het-FH	9 countries ***	306	LLTΩ & anacetrapib 100 mg for 12 months	Anacetrapib is substantially reducing LDL-C across all genotypes	Kastelein et al., 2016 [80]
*LDLR*	Defective and negative *LDLR*	Hom-FH	American	8	Statins, ezetimibe, mipomersen, lomitapide, PCSK9 inhibitors, & gemcabene 300, 600 or 900 mg	Gemcabene is reducing LDL-C in uncontrolled cases under LLT treatment	Gaudet et al., 2019 [81]
*LDLR PCSK9 APOB LDLRAP1*	Pathogenic causative Gain-of-function Pathogenic Pathogenic	Het-FH	8 countries **	432	LLTΩ & inclisiran 300 mg/3 months	Inclisiran is significantly reducing LDL- C in *LDLR* variants	Raal et al., 2020 [82]
*LDLR* *APOB*	Pathogenic	Het-FH	Various countries **	1887	LLTΩ, evolocumab or alirocumab, & inclisiran	PCSK9 inhibitors is reducing LDL-C among all genetic mutations	Brandts et al., 2021 [83]

** Various sites in Europe, Africa, the Middle East, North America, and South Africa. *** The 9 countries include the Netherlands, France, Spain, Canada, the USA, Germany, Russia, Norway, and the UK. LLT including ezetimibe, fibrates, statins, lomitapide, PCSK9 inhibitors, bile acid sequestrants (e.g., cholestyramine and colestipol), or niacin. Abbreviations: LLT, lipid-lowering therapies; FH, familial hypercholesterolemia; Het-FH, patients with heterozygous FH; Hom-FH, patients with homozygous FH; ApoB, Apolipoprotein B; HDL-C, High-density lipoprotein cholesterol; LDL-C, low-density lipoprotein-C; TC, total cholesterol; TG, triglyceride; LDLR, Low-density lipoprotein receptor; APOB, Apolipoprotein B; HMGCR, β-hydroxy-β-methylglutaryl Coenzyme A Reductase; LDLRAP1, LDLR-adaptor protein 1; PCSK9, proprotein convertase subtilin/kexin 9 genes. * Borderline significance (*p* < 0.05).

Ezetimibe is principally metabolized by the intestinal and hepatic enzymes, uridine 5′-diphosphate (UDP)-glucuronosyltransferase (UGT) and excreted in the urine (Figure 2) [6]. So far, genetic analysis has not presented any influence of pharmacokinetic genes on this drug.

### 4.2. Monoclonal Antibodies to PCSK9

PCSK9 enzyme is quintessential for recycling LDLR and eliminating lipoproteins from the bloodstream. The functional research recognized that decreased cholesterols and heart diseases, as well as a more extraordinary response to statins, have been presented in loss-of-function mutation carriers [55]. However, a gain of function variant (functional SNP) was linked to low LDLR expression and statins resistance [56]. Based on this genomic discovery, PCSK9 inhibitors were developed and immediately became a target for the clinical management of FH. Evolocumab, alirocumab, and inclisiran are the approved anti-PCSK9 monoclonal antibodies as an additive therapy to the aggressive treatment regimen of FH patients. These medications inhibit the PCSK9 binding with LDLR and, thus, enhance hepatic LDLR expression and reduce the circulating lipoproteins. In the mild FH phenotype, evolocumab 140–420 mg subcutaneously every 2–4 weeks raises the LDL-C reduction by 54–60%, respectively. Alirocumab 75 or 150 mg subcutaneously every two weeks has also decreased the levels of LDL-C, total cholesterol, and ApoB in heterozygous subjects by 51–58% [72]. Interestingly, the response to PCSK9 inhibitors is influenced by the baseline mutations in homozygous and heterozygous FH individuals.

Different responses to anti-PCSK9 monoclonal antibodies have been reported with superior sensitivity to alirocumab compared with evolocumab. This differential efficacy was found in patients with heterozygous FH and those at high CVD risk and resistance to statins [67,72]. Blom and colleagues recently demonstrated that the combination of alirocumab with classical therapy in homozygous cases carrying double *LDLR* allele leads to notable regulating of the plasma lipids [78]. Conversely, the optimizing of low LDL-C is hardly obtained with evolocumab treatment in homozygous FH patients carrying nonfunctional *LDLR* due to the *LDLR*-dependent mechanism of such agents [66]. Numerous analyses have concluded that the pharmacological effect of evolocumab is based on the phenotype-genotype mutation of *LDLR*. They found that subjects carrying defective *LDLR* alleles are highly sensitive to treatment and those with an autosomal recessive FH are moderately sensitive. At the same time, individuals with no *LDLR* function (receptor-negative mutations) do not respond to evolocumab [15,65,81]. Generally, the therapeutic efficacy of evolocumab was found to be dependent on various phenotypes.

The *LDLRAP1* genotype (c.1A > G) was associated with an attenuated response of autosomal recessive FH patients to evolocumab [74]. Reciprocally, a higher reduction of LDL-C was observed by evolocumab in patients carrying another *LDLRAP1* variant (c.136 C > T (406)) with resistance to traditional medications [70]. This observation disproves the fact that evolocumab would not demonstrate an effective response in patients with *LDLRAP1* variants. Patients with homozygous FH resulted from gain-of-function missense variants in *PCSK9,* and two mutant alleles of *LDLR* genes might have a worse phenotype with negligible response to anti-PCSK9 antibodies and statins [48,76]. Compared to heterozygous FH subjects with typical *LDLR* mutations, those with a gain-of-function variant, D374Y *PCSK9*, havda more aggressive phenotype with excessive lipid levels, risk of CVD, and poor sensitivity to lipid-neutralizing medicines [84]. This indicates that the intensity of FH depends on the functional genetic mutation in addition to the number of defected alleles, homozygosity, and heterozygosity.

The phase 3 ORION pilot studies manifested that inclisiran 300 mg twice a year could robustly minimize cholesterol concentrations by 50% and PCSK9 by 70% in FH patients with heart defects. Inclisiran 300 mg subcutaneously is a novel and the only approved small interfering RNA (siRNA) agent that selectively inhibits the hepatocyte synthesis of PCSK9 [6]. Remarkably, a greater LDL-C reduction was observed after the second dose administration compared to only a single dose with an acceptable safety profile. The suspended therapeutic effect of siRNA with a low administration frequency is a unique advantage of inclisiran over other adjunct anti-lipids. It provides a long-term adherence that minimizes CVD events in patients at high risk. Raal et al. recognized that inclisiran powerfully and safely decreases the cholesterol levels among all cases of mild FH genotype carrying disease-causative polymorphisms, including *LDLR, APOB, PCSK9,* and *LDLRAP1* [82]. A very recent meta-analysis study in heterozygous FH subjects carrying different phenotypes has observed similar physiological effects of anti-lipids targeting PCSK9 [83].

PCSK9 inhibitors are known to be eliminated through the intestinal pathway and bypassing hepatic metabolism (Figure 2). Remarkably, kinetic studies confirmed that negative *APOB* carriers have a lower serum concentration of alirocumab than those with *PCSK9* gain-of-function variants [85]. In support of this, another genetic analysis proved a significant therapeutic response to anti-PCSK9 antibodies in FH patients with *APOB* variants (rs5742904) [69]. Another antibody against PCSK9, bococizumab 150 mg subcutaneously every other week, was characterized by a weak nontoxic profile along with a short-term attenuation of LDL-C due to the great neutralizing force of the defending antibodies [50]. Considering the cost-benefit analysis of this medication, it can possibly be used in FH cases at high risk of CVD [6]. The genomic examination of FH patients at risk for *PCSK9/LDLR* and *APOB* polymorphisms has become necessary to ameliorate clinical diagnosis and management by considering the use of PCSK9 inhibitors in their therapeutic care plan.

### 4.3. Mipomersen

Mipomersen 200 mg subcutaneously per week is recommended as an adjunct to standard anti-lipid therapy with a low-fat diet in homozygous FH patients. A second-generation antisense oligonucleotide (ASO) lowers cholesterol through selective degradation of hepatic ApoB-100 messenger ribonucleic acid (mRNA) transcript. This ultimately leads to a sustained reduction of atherogenic ApoB-100 containing lipoproteins, including lipoprotein, VLDL, and LDL-C, via an LDLR-independent pathway. This pathway was targeted because rare *APOB* polymorphisms are one of the causative factors of FH. Although the FDA has approved the use of mipomercen in FH patients, the European Committee for Medicinal Products for Humans has terminated mipomersen due to its life-threatening hepatotoxicity [6].

Interestingly, mipomersen has a wide interindividual variability in controlling lipid levels among homozygous and heterozygous individuals. Mipomercin was associated with a 21% decrease in LDL-C, a 25% reduction in lipoproteins, and a 22% decrease in ApoB levels in patients with heterozygous FH [86]. In homozygous cases, the combination of mipomersen and standard lipid-lowering therapy was accompanied by ApoB reduction of 46% and LDL-C reduction of 42% [87]. The metabolism of mipomersen does not depend on conventional drug-metabolizing enzymes, and thus does not interact with concomitant agents and is excreted mainly through the urinary pathway [86]. In general, mipomersen could significantly reduce ApoB and LDL-C but with limited tolerability and variable effect in FH patients.

### 4.4. Lomitapide

The MTP protein plays a substantial role in VLDL and chylomicron’s hepatic and intestinal assembly, respectively. Loss-of-function mutations in MTP result in limited plasma levels of ApoB-48 and ApoB-100 in addition to hypocholesterolemia [88]. Another examination confirmed that MTP -493 GT SNP has a gender-specific restriction of atorvastatin-induced lipid reduction [38]. This suggests targeting *MTP* to manage hypercholesterolemia. Lomitapide 5–60 mg orally per day is the only approved MTP inhibitor to treat patients with homozygous FH. Severe defect in LDLR and CYP3A4 function attenuates the drug efficiency that targets LDL-C elimination [6]. It inhibits the secretion of lipoproteins into the bloodstream and reduces the LDL-C by 38% combined in homozygous FH patients. D’Erasmo and colleagues illustrated that a combination of lomitapide with traditional medicines in cases with the severe FH phenotype had been correlated with a very efficient and well-tolerated lipid reduction [68]. In India, it was found that using a PCSK9 inhibitor, evolocumab 420 mg every month, combined with standard therapy in homozygous FH patients carrying impaired *LDLR* activity was ineffective in controlling plasma lipids or limiting the number of heart diseases. The addition of lomitapide powerfully reduced 54% of the LDL-C and 15% of major coronary artery diseases [77]. Thus, utilizing lomitapide as adjunct therapy can potentially and safely optimize the reduction of LDL-C through genotype-independent effects in FH subjects [73,79].

## 5. Pharmacogenomics of Novel Lipid-Lowering Therapies in FH

Based on the knowledge of pathological genetic mutations involved in the intrinsic or extrinsic cholesterol pathways, therapeutic research has discovered novel strategies with unique mechanisms that substantially enhance the management of dyslipidemia. Gene-based medicines are categorized into integrated genomic replacement treatment that inserts healthy genes to replace pathological mutants, modification of gene expression, and transcription that target coding or noncoding RNAs to alter singling or splicing mechanisms and, ultimately genome modification to insert or delete a specific genetic sequence [2]. Gene therapy showed potent and persistent reduction of LDL-C and elevation of *LDLR* expression in homozygous FH by restoring the functional hepatic LDL-C elimination [89].

Numerous emerging or new pharmacological approaches are designated to target functional genes for the management of unresponsive or severe FH (Table 2 and Figure 1). However, little is known about the efficiency and resistance of such strategies among FH patients with different genotypes.

### 5.1. Evinacumab

Loss of function mutations in hepatic angiopoietin-like protein 3 (ANGPTL3) results in low levels of LDL-C, high-density lipoprotein cholesterol (HDL-C; good cholesterol), and triglyceride. Evinacumab 15 mg/kg intravenously every month is a new monoclonal antibody treatment targeting the ANGPTL3 protein, an endogenous lipoprotein lipase inhibitor [6]. Importantly, this inhibitory mechanism leads to well-tolerated and powerful triglyceride depletion by 50%, HDL-C by 30%, and LDL-C by 47% via bypassing the LDLR expression [90]. Multiple investigations have confirmed that evinacumab can effectively optimize a minimal level of LDL-C in patients with homozygous and heterozygous FH independently of *LDLR* mutations [71,75]. This provides a highly targeted approach to treat individuals with *LDLR* impairments who are resistant to other anti-lipids, such as PCSK9 and HMGCR inhibitors. The ANGPTL3 inhibitor was recently approved to be prescribed on top of an aggressive lipid-lowering treatment for homozygous FH pediatric patients of 12 years of age or more depending on the phase 3 ELIPSE trial [90].

### 5.2. Bempedoic Acid

Bempedoic acid 180 mg by oral daily is another newly approved cholesterol-lowering treatment for FH subjects with CVD and statin intolerance. It is a robust adenosine triphosphate citrate lyase (ACL) inhibitor and an activator of AMP-activated protein kinase (AMPK) in the liver. This ACL inhibitor is an inactive agent that is activated through the metabolic activity of a very-long-chain acyl-CoA synthetase-1 (ACSVL1), and then deactivates via UGT hepatic enzymes. The direct mechanism of bempedoic acid is to restrict cholesterol and fatty acid production, thus upregulating hepatic LDLR and depleting cholesterol, inflammatory C-reactive protein, and LDL-C [6]. The combination of bempedoic acid along with atorvastatin and ezetimibe has been associated with a fundamental and long-term reduction of cholesterol by nearly 50% and C-reactive protein by 40% across FH patients at high risk of ASCVD with no major toxicities [91]. This ACL inhibitor is an inactive agent that is activated through the metabolic activity of very-long-chain acyl-CoA synthetase-1 (ACSVL1) and then deactivated via UGT hepatic enzymes.

### 5.3. Gemcabene

A novel lipid-regulating mechanism has been established in gemcabene which promotes apolipoprotein molecule degradation through decreasing the messenger RNA of apolipoprotein C-III (ApoC-III) in the liver. Up to the present time, gemcabene 450 to 900 mg orally a day has been found to be effective and well-tolerated among many different patient groups for three months. It can exceedingly diminish ApoB, C-reactive protein, and LDL-C by 30%, as well as raise HDL-C in FH patients on top of optimal therapy independently of LDLR. Importantly, gemcabene effectively reduced LDL-C levels by 44% in homozygous FH patients with negative-*LDLR* mutations [81]. This indicates that gemcabene could be used in patients with nonfunctional LDLR that are resistant to statins and PCSK9 inhibitors.

### 5.4. CETP Inhibitor

Cholesteryl ester transfer protein (CETP) is responsible for the heteroexchange between atherogenic ApoB-lipoproteins, particularly VLDL, and HDL-C of triglycerides and cholesteryl esters. Distinctively, it is characterized by a long-acting kinetic effect caused by the increased adipose tissue accumulation. The lack of CETP activity caused by genetic defects was accompanied by low LDL-C levels and a consequent CVD risk, as well as elevated HDL-C. Anacetrapib, a new direct inhibitor of CEPT, was analyzed in a large cohort cardiovascular study. A substantial 9% reduction of major CVD accompanied by nearly 30% reduction of cholesterols was reported in heterozygous FH cases [92]. Nevertheless, despite the acceptable nontoxic profile, the sponsor decided to discontinue the anacetrapib commercialization and has not proposed that it get clinical approval. A global study was conducted on a large population of heterozygous FH patients who have been treated with anacetrapib 100 mg orally a day for up to 52 weeks as adjunctive therapy to optimal anti-lipids. It found an almost 40% decrease in LDL-C [80]. The cholesterol reduction observed in these cases holds future promise in terms of managing cases with unmanaged FH who are resistant to the most aggressive therapies.

## 6. Conclusions and Clinical Prospect of the Future

The overburden of prolonged hypercholesterolemia increases the incidence of life-threatening consequences such as myocardial infarction, especially in FH patients who are generally undiagnosed and uncontrolled. Despite the improvements in lipid-neutralizing therapies, several genetic and non-genetic factors may greatly influence the pharmacodynamic and pharmacokinetic pathways. Throughout the past decade there has been an unprecedented development in the study of genetic variants. Emerging approaches to pharmacogenetic analysis have extended the clinical surveillance of novel candidate genotypes and phenotypes, improving our knowledge of the biochemical effect of anti-lipids and the impact of genetic variations on clinical outcomes. As a result, various new anti-lipids have been discovered, depending on the discovered novel and rare mutations in addition to the genetic pathophysiology of diverse rare diseases, including FH. However, pharmacogenomics’ lack of appropriate medical implications has drastically impacted the optimal treatment of many pathologies.

Ideally, future pharmacogenomic analysis of lipid-regulating agents should focus on including various ethnic backgrounds as well as on understanding and comparing the impact of genetic/epigenetic variants on the anti-lipid’s physiological pathways. The exploitation of GWAS results for ethnic groups is needed to promote medical outcomes and prevent major complications, such as ASCVD, for FH or dyslipidemia patients. Therefore, whole-genome sequencing can contribute significantly to the personalization of FH therapeutic regimens based on the patient’s complete genetic profile. Consequently, we proposed the strategy of diagnosing and managing patients with FH and their families according to current guidelines as illustrated in Figure 3 [6]. We strongly recommend genomic screening for patient-specific variants prior to treatment, particularly for subjects with major pathogenic polymorphisms. Furthermore, patients and their families should be counseled about the advantages of detecting the disease-causative gene mutations as well as utilizing novel anti-lipids such as evinacumab, inclisiran, gemcabene, and anacetrapib in severe and unresponsive FH cases. Ultimately, regular clinical follow-up is strongly recommended in our strategy to determine interindividual variability of therapeutic outcomes among patients of different genotypes. If applied appropriately, this gene-based, personalized medicine and evaluation will help to promote drug potency, tolerability, and safety as well as to sustain a healthy quality of life in patients with hereditary diseases.

## Figures and Tables

**Figure 1 jpm-11-00877-f001:**
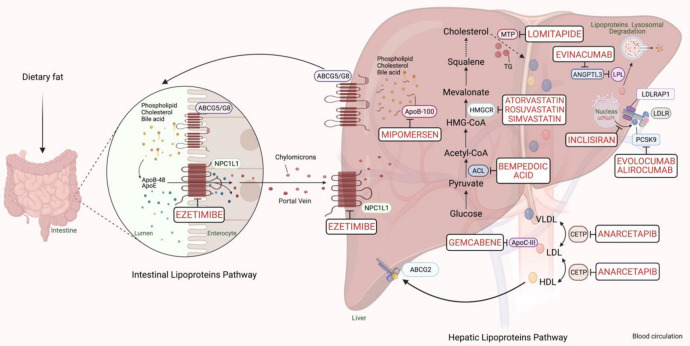
Scheme representing the cholesterol metabolism pathway and pharmacological targets for classical and novel lipid-lowering therapies (generated with BioRender.com). Unique hepatic and intestinal transporters (ABCG5 and ABCG8) release bile acids, phospholipids, and cholesterol into the biliary system. Inversely, the NPC1L1 protein modulates cholesterol returning to hepatocytes. Ezetimibe inhibits cholesterol entry into the intestine and liver by the NPC1L1 transporter. Next, chylomicrons are generated through the assembly of TG, cholesterol, and ApoB-48, and released into the blood circulation (see intestinal lipoproteins pathway). The speed-limiting enzyme of endogenous cholesterol synthesis, HMGCR (see hepatic lipoproteins pathway), is inhibited via statins. The activated bempedoic acid decreases the hepatic synthesis of acetyl CoA and cholesterol catabolism by blocking the ACL protein. ApoB-100, phospholipid, ApoC-III, and cholesterol assembly into VLDL depends on the activity of MTP, which is blocked by lomitapide. The degradation of hepatic messenger ribonucleic acid (mRNA) transcript of ApoB-100 and ApoC-III is mediated by mipomersen and gemcabene, respectively. The heteroexchange of TGs and cholesteryl esters between ApoB-lipoproteins particles relies on CETP activity, which is blocked by CETP inhibitors such as anacetrapib. LDLR interacts and removes LDL-cholesterol from the blood circulation with the assistance of LDLRAP1. Lysosomal catabolism of LDLR is mediated by PCSK9. Anti-PCSK9 antibodies, including evolocumab, alirocumab, and inclisiran inhibit the endogenous production and release of PCSK9. Evinacumab blocks the inhibition of hepatic lipoprotein lipase activity by ANGPTL3. Abbreviations: ApoB-48/100, Apolipoprotein B protein member 48 & 100; ApoC-III, Apolipoprotein C protein, member III; ApoE, Apolipoprotein E protein; HDL, High-density lipoprotein cholesterol; LDL, low-density lipoprotein; VLDL, very-low-density lipoprotein; TG, triglyceride; LDLR, LDL-receptor; LDLRAP1, LDLR-adaptor protein, member 1; ABCG2, atp-binding cassette, subfamily g, member 2; HMGCR, β-hydroxy-β-methylglutaryl Coenzyme A Reductase; NPC1L1, Niemann-Pick C1-like 1 transporter protein; ACL, adenosine triphosphate citrate lyase; MTP, microsomal triglyceride transfer protein; LPL, lipoprotein lipase; CETP, cholesteryl ester transfer protein; ANGPTL3, angiopoietin-like protein 3; PCSK9, proprotein convertase subtilin/kexin 9 protein.

**Figure 2 jpm-11-00877-f002:**
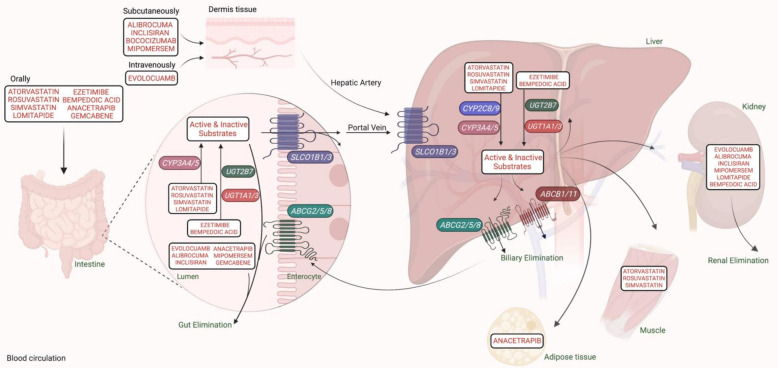
A schematic representation of the pharmacokinetic modulator genes involved in absorption, distribution, metabolism, and clearance of lipid-lowering therapy (created with BioRender.com). Oral lipid-antagonists enter the circulatory system via the enteric SLC and ABC gene-transporters. While intravenous anti-lipids enter directly into the circulation and reach the liver, the agents administrated subcutaneously are slowly absorbed through the blood capillaries. The liver and kidneys are the major metabolic sites for lipid-lowering medicines. The main catalytic proteins involved in their metabolic pathway are CYP and UGT, which inactivate or activate drugs. Members of the ABC family then mediate their elimination through kidneys, biliary, or intestinal pathways. Some drugs accumulate for a long-time in the muscle or adipose tissue. Abbreviations: ABCG2/5/8, atp-binding cassette, subfamily g, member 2, 5, or 8; SLCO1B1, solute carrier organic anion transporter 1B1; CYP3A4, Cytochrome P450, family 3, subfamily A, member 4; UGT2B7, uridine 5′-diphosphate (UDP)-glucuronosyltransferase 2B7; UGT1A1/3, uridine 5′-diphosphate (UDP)-glucuronosyltransferase 1A1 or 3.

**Figure 3 jpm-11-00877-f003:**
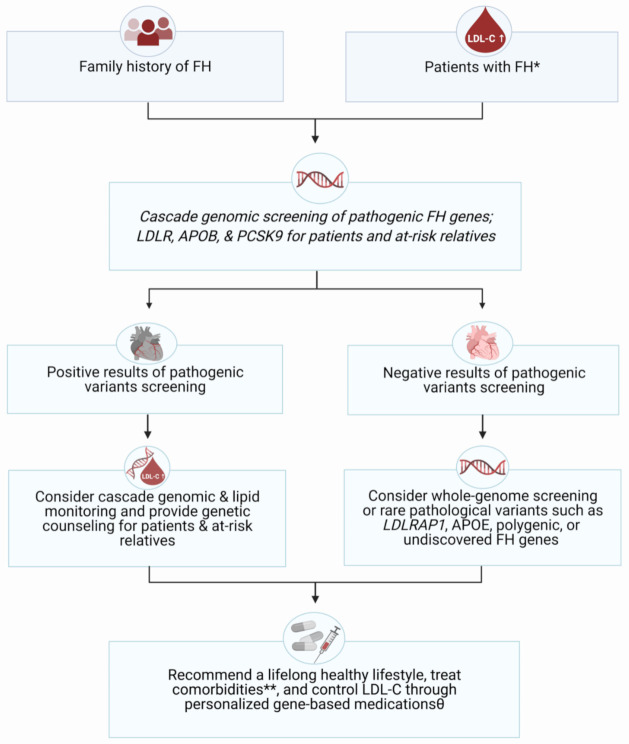
Flowchart illustration of the recommended genomic screening process for different groups of FH patients and their families (generated with BioRender.com). * Diagnostic Criteria of FH based on Dutch-MEDPED guideline: total cholesterol > 250 mg/dL, LDL-C > 190 mg/dL (adults) or >160 mg/dL (children), in addition to family history of similar findings or with premature cardiovascular diseases, tendon xanthomas, arcus cornealis, or DNA-based evidence of *LDLR, APOB*, or *PCSK9* functionality mutations [6]. ** Morbidities associated with FH: cardiovascular diseases such as coronary heart disease, stroke, & peripheral vascular disease, diabetes, hypertension, and erectile dysfunction. Θ Regularly monitor and compare the response and safety of medications according to each individual genotypes. Abbreviations: LDL-C, low-density lipoprotein-cholesterol; *LDLR*, LDL-receptor; *APOB,* Apolipoprotein B; *PCSK9*, proprotein convertase subtilin/kexin 9 protein, *LDLRAP1*, LDLR-adaptor protein, member 1, *APOE,* Apolipoprotein E.

## Data Availability

Not applicable.

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
