# Peer review of "Pharmacogenomics Variability of Lipid-Lowering Therapies in Familial Hypercholesterolemia"

_jpm, 2021, doi:10.3390/jpm11090877_

Round 1
Reviewer 1 Report
The authors have comprehensively reviewed the literature on the management of familial hypercholesterolemia in the view of pharmacogenomics variability of lipid-lowering therapies. Pharmacogenomic analysis may have crucial clinical implications in the future. The manuscript is valuable and clearly written.
Minor linguistic concerns are listed below:
- Description of Figure 1 (line 139) "Lysosomal catabolism of LDLR mediates by PCSK9". I suggest to write "is mediated" instead
- Line 179-180 The sentence should be revised. The use of "although" and "however" in the same sentence does not seem to be necessary.
- Table 1 - Null and defective LDLR mutation (Heath et al.) - the formulation "LDL-C reduction is more in patients" should be replaced with "LDL-C reduction is higher" as in other lines
- Table 1 - PCSK9 E32K mutation - "PCSK9 gain of function variants significantly worse the LDLR" please correct it to "worsen"
- I suggest to change "HDL" abbreviation into "HDL-C" as the authors did in case of LDL-C
- Abbreviations - Table 1 - "Tg" is not bolded
- Table 2 - "LDLR and LDLRAP1 variants reduces" please replace it with "reduce"
- Table 2 - "Lomitapide powerfully reduced lipid profile" (Mahzari et al." - in every line authors use Present Simple Sentences, I suggest to keep verb tense consistency.
- Line 394 Please replace" have a worsen phenotype" with "have a worse phenotype"
- Please add author contribution, acknowledgments and conflict of interests
Author Response
Comments and Suggestions for Authors from Reviewer 1:
“The authors have comprehensively reviewed the literature on the management of familial hypercholesterolemia in the view of pharmacogenomics variability of lipid-lowering therapies. Pharmacogenomic analysis may have crucial clinical implications in the future. The manuscript is valuable and clearly written.”
We thank reviewer 1 for her/his valuable assessment of our manuscript and we have incorporated all the suggested corrections and changes accordingly (see below) in the newly uploaded version.
Minor linguistic concerns are listed below:
- Description of Figure 1 (line 139) "Lysosomal catabolism of LDLR mediates by PCSK9". I suggest to write "is mediated" instead.
- We changed this sentence accordingly.
- Line 179-180 The sentence should be revised. The use of "although" and "however" in the same sentence does not seem to be necessary.
- We changed this sentence as suggested, “Although statins robustly diminish cholesterol in addition to CVD morbidity and mortality by 20-30% in normal individuals, their efficacy is predominantly weaker in FH subjects” (line 182-183).
- Table 1 - Null and defective LDLR mutation (Heath et al.) - the formulation "LDL-C reduction is more in patients" should be replaced with "LDL-C reduction is higher" as in other lines.
- We changed this sentence accordingly.
- Table 1 - PCSK9 E32K mutation - "PCSK9 gain of function variants significantly worse the LDLR" please correct it to "worsen".
- We changed this sentence accordingly.
- I suggest to change "HDL" abbreviation into "HDL-C" as the authors did in case of LDL-C.
- We changed this abbreviation accordingly.
- Abbreviations - Table 1 - "Tg" is not bolded.
- We changed this abbreviation to bold style.
- Table 2 - "LDLR and LDLRAP1 variants reduces" please replace it with "reduce".
- We changed this sentence accordingly.
- Table 2 - "Lomitapide powerfully reduced lipid profile" (Mahzari et al." - in every line authors use Present Simple Sentences, I suggest to keep verb tense consistency.
- We changed all verbs to the same tense as suggested in all the tables.
- Line 394 Please replace" have a worsen phenotype" with "have a worse phenotype".
- We changed this sentence accordingly.
- Please add author contribution, acknowledgments and conflict of interests.
- We declared the authors contribution, acknowledgments, and conflict of interests.
Reviewer 2 Report
The authors provided a literature review on Pharmacogenomics Variability of Lipid-Lowering Therapies in Familial Hypercholesterolemia. The review describes a large number of articles that cover almost all lipid-lowering drugs. I have no major concern, however, the article section "Conclusion and Clinical Prospect of the Future" can be improved by adding clearer practical recommendations that will help make better use of the information provided. An explanation of the term "cardiomyopathies" is recommended (line 79, line 525)
Author Response
Comments and Suggestions for Authors from Reviewer 2:
The authors provided a literature review on Pharmacogenomics Variability of Lipid-Lowering Therapies in Familial Hypercholesterolemia. The review describes a large number of articles that cover almost all lipid-lowering drugs. I have no major concern, however, the article section "Conclusion and Clinical Prospect of the Future" can be improved by adding clearer practical recommendations that will help make better use of the information provided. An explanation of the term "cardiomyopathies" is recommended (line 79, line 525).
We thank reviewer 2 for his/her positive assessment of the manuscript and we have carried on the necessary modifications as suggested (see below) in the newly revised version.
- We explained the meaning and pathophysiology of cardiomyopathy as recommended “coronary artery disease and heart attacks restrict coronary blood flow, causing the pumping chamber to enlarge, widen, and attenuate. Ultimately, this damage will lead to ischemic cardiomyopathy, potentially reducing the ability of cardiomyocytes to pump blood” (line 90-93).
- As recommended by the reviewer, we adjusted the “Conclusion and Clinical Prospect of the Future” as follows: we recommended a genomics-based clinical strategy (line 559-580) that can improve early prediction, diagnosis, and management as well as awareness of patients and their families about FH, illustrated in a simple flowchart-based on the recently published Dutch-MEDPED FH guidelines (Figure 3).